EMBO
Molecular Medicine

# Mutations in pregnancy-associated plasma protein A2 cause short stature due to low IGF-I availability

Andrew Dauber[1], María T Muñoz-Calvo[2,3], Vicente Barrios[2,3], Horacio M Domené[4], Soren Kloverpris[5], Clara Serra-Juhé[6], Vardhini Desikan[7], Jesús Pozo[2,3], Radhika Muzumdar[8], Gabriel Á Martos-Moreno[2,3], Federico Hawkins[9], Héctor G Jasper[4], Cheryl A Conover[10], Jan Frystyk[11,12], Shoshana Yakar[13], Vivian Hwa[1], Julie A Chowen[2,3], Claus Oxvig[5], Ron G Rosenfeld[14,15], Luis A Pérez-Jurado[6] & Jesús Argente[2,3,*]

## Abstract

Mutations in multiple genes of the growth hormone/IGF-I axis have been identified in syndromes marked by growth failure. However, no pathogenic human mutations have been reported in the six high-affinity IGF-binding proteins (IGFBPs) or their regulators, such as the metalloproteinase pregnancy-associated plasma protein A2 (PAPP-A2) that is hypothesized to increase IGF-I bioactivity by specific proteolytic cleavage of IGFBP-3 and -5. Multiple members of two unrelated families presented with progressive growth failure, moderate microcephaly, thin long bones, mildly decreased bone density and elevated circulating total IGF-I, IGFBP-3, and -5, acid labile subunit, and IGF-II concentrations. Two different homozygous mutations in *PAPPA2*, p.D643fs25* and p.Ala1033Val, were associated with this novel syndrome of growth failure. *In vitro* analysis of IGFBP cleavage demonstrated that both mutations cause a complete absence of PAPP-A2 proteolytic activity. Size-exclusion chromatography showed a significant increase in IGF-I bound in its ternary complex. Free IGF-I concentrations were decreased. These patients provide important insights into the regulation of longitudinal growth in humans, documenting the critical role of PAPP-A2 in releasing IGF-I from its BPs.

**Keywords** bone; delayed growth; growth hormone; IGF-binding proteins; IGF bioavailability
**Subject Categories** Genetics, Gene Therapy & Genetic Disease

## Introduction

The growth hormone (GH)/insulin-like growth factor (IGF)-I system is essential for optimal human growth. GH promotes growth via IGF-I production and by direct actions on the growth plate. IGF-I circulates bound to six IGF-binding proteins (IGFBPs). After binding IGF, IGFBP-3, and -5 also bind with the IGF acid labile subunit (ALS) to form a ternary complex, which further increases IGF-I's half-life (Baxter, 2000). Free IGF-I (fIGF-I) binds its receptor, activating signaling cascades that up-regulate multiple genes fundamental to growth (Yakar *et al*, 2002). Human genetic defects in this axis lead to syndromes marked by impaired growth and have helped to further our understanding of growth physiology (David *et al*, 2011). GH receptor (GHR) mutations were shown to cause Laron syndrome with extreme growth failure (Amselem *et al*, 1989). Mutations in the *STAT5B* (Kofoed *et al*, 2003), IGF-I (Woods *et al*, 1996) and IGF-I receptor (IGF-IR) (Abuzzahab *et al*, 2003) genes cause varying degrees of pre- and postnatal growth retardation and mutations in ALS cause mild short stature (Domene *et al*, 2004). The first mutation in *IGF2* affecting pre- and postnatal growth was recently identified (Begemann *et al*, 2015). To date, no human mutations in

1  Cincinnati Center for Growth Disorders, Division of Endocrinology, Cincinnati Children's Hospital Medical Center, Cincinnati, OH, USA
2  Department of Pediatrics & Pediatric Endocrinology, Hospital Infantil Universitario Niño Jesús, Instituto de Investigación La Princesa, Universidad Autónoma de Madrid, Madrid, Spain
3  Program of Pediatric Obesity, CIBEROBN, Instituto de Salud Carlos III, Madrid, Spain
4  Centro de Investigaciones Endocrinológicas "Dr. César Bergadá" (CEDIE), CONICET, FEI, División de Endocrinología, Hospital de Niños Ricardo Gutiérrez, Buenos Aires, Argentina
5  Department of Molecular Biology and Genetics, Aarhus University, Aarhus, Denmark
6  Genetics Unit, Universitat Pompeu Fabra, Hospital del Mar Research Institute (IMIM) & CIBERER. Instituto de Salud Carlos III, Barcelona, Spain
7  Department of Pediatrics, Division of Pediatric Endocrinology, New York Medical College, Valhalla, NY, USA
8  Division of Endocrinology, Children's Hospital of Pittsburgh, Pittsburgh, PA, USA
9  Department of Endocrinology, Hospital Universitario 12 de Octubre, Universidad Complutense de Madrid, Madrid, Spain
10  Division of Endocrinology, Mayo Clinic, Rochester, MN, USA
11  Medical Research Laboratory, Department of Clinical Medicine, Faculty of Health, Aarhus University, Aarhus, Denmark
12  Department of Endocrinology and Internal Medicine, Aarhus University Hospital, Aarhus, Denmark
13  Department of Basic Science and Craniofacial Biology, New York University College of Dentistry, New York, NY, USA
14  Oregon Health and Science University, Portland, OR, USA
15  STAT5 LLC, Los Altos, CA, USA
*Corresponding author. Tel: +34 91 5035912; Fax: +34 91 5035939; E-mail: jesus.argente@uam.es

the six high-affinity IGFBPs or their regulatory proteins have been reported in association with a monogenic syndrome. We present two families with homozygous loss-of-function mutations in *PAPPA2*, a gene encoding pregnancy-associated plasma protein-A2 (PAPP-A2), a protease highly specific for IGFBP-3 and -5 (Overgaard *et al*, 2001), resulting in a novel syndrome of growth retardation with markedly elevated circulating IGF-I and IGF-II, but decreased fIGF-I concentrations and IGF bioactivity.

# Results

### Patient descriptions

Family one consists of four children of Spanish ancestry born to parents with no known consanguinity. Subject II.1 was evaluated at nine years of age for short stature with a height 1.7 SDS below her mid-parental target height (50th–75th percentile) and elevated serum IGF-I, ALS, and IGFBP-3 concentrations (Table 1). Spontaneous GH secretion over eight hours (daytime) was elevated (4.8 ng/ml/8 h; normal 2.8–3.3). Her younger brother's height (Subject ll.3) was 1.3 SDS below mid-parental target height, with similar elevations in IGF-I, IGFBP-3, and ALS (Table 1) and spontaneous GH secretion (5.3 ng/ml/8 h). Their birth lengths and weights were normal, although smaller than their unaffected sisters (II.2 and II.4; Table 1). Subject II.3 was born premature, but was adequate for gestational age, with a normal catch-up growth (Fig 1B). Their growth patterns revealed consistent short stature relative to target height, with more prominent growth deceleration as they age (Fig 1, Appendix Fig S1). Subtle dysmorphic features consist of small chins, mild microcephaly (−1.37 & −1.65 SDS at birth; −2.03 & −2.19 SDS at first physical examination), and long fingers and toes (Fig 1A and B). They are prepubertal. Glycemia was normal on oral glucose tolerance tests, but with mild fasting hyperinsulinemia (15.1 and 27 μU/ml, normal 4–13). The unaffected sisters were within 1 SDS of their mid-parental target height with normal IGF-I serum concentrations (Table 1, Appendix Fig S2).

Family two consists of five children of Palestinian ancestry born to parents of normal stature (mid-parental target height 30th percentile) who are first cousins. Siblings II.3, II.4, and II.5 had significant postnatal growth retardation with high serum IGF-I and IGFBP-3 concentrations. When Subject II.3 was evaluated for short stature at age 6.8 years of age, her height was −3.5 SDS below her target height and IGF-I was 517 ng/ml (normal 47–217). An elevated IGFBP-3 level of 6,400 ng/ml (normal 2,100–4,200) was found later. She showed peak GH concentrations > 50 ng/ml on both clonidine and arginine tests. Her growth chart demonstrates progressive growth failure, no discernible pubertal growth spurt and a final adult height of 138.4 cm (−3.81 SDS; Fig 1C). She had normal pubertal timing, undergoing menarche at age 13.8 years of age, and had elevated IGF-I throughout childhood (Appendix Table S1; most recent value 1,060 ng/ml at age 18 years). Two younger brothers also had progressive growth failure (Fig 1D and E). Subject II.4 was evaluated for growth failure at age 12.5 years of age with an IGF-I level of 657 ng/ml (normal 93–567) despite being prepubertal. Subject II.5 was evaluated at 8.3 years of age with an IGF-I level of 636 ng/ml (normal 49–351). Their peak GH concentrations after stimulation testing with clonidine and L-Dopa were 14.7 and

37 ng/ml, respectively. Fasting glucose concentrations were normal in both brothers, with mild hyperinsulinemia in Subject II.4 (17.1 μU/ml) and a normal insulin level of 7 μU/ml in Subject II.5. Subjects II.3 and II.4 were born mildly small for gestational age, but Subject II.5 was within the normal range (Table 1). All three affected siblings had long thin fingers, a small chin, moderate microcephaly (Table 1) and delayed dental eruption.

Skeletal surveys in the two affected siblings from family one and subjects II.4 and II.5 in family two showed no signs of overt skeletal dysplasia; however, thin long bones most prominent in the fibulae, tibiae, and femurs were found (Fig 1F). Bone age was consistent with chronologic age in all subjects.

DXA scans (QDR 4500, Hologic, Waltham, MA) were performed on the two affected siblings from family one. Bone mineral density was decreased at the lumbar spine (height adjusted Z-score −2.49 SDS in II.1, −2.0 SDS in II.3).

Micro-CT analysis of a tooth extracted from Subject II.1 in family one showed significantly decreased enamel and dentin density (Fig EV1). No significant cognitive dysfunction or hearing abnormalities were found in the patients from either family.

### Genetic analysis

Given the two affected siblings of different sex in family one with unaffected parents, we hypothesized that the disorder had an autosomal recessive inheritance pattern. After excluding mutations in the IGF-IR gene, whole-exome sequencing was performed in the affected sister (II.1). We searched for genes with either a single homozygous or two heterozygous rare (minor allele frequency < 1%) nonsynonymous variants present. Five genes met these criteria (Appendix Table S2), four of which fell within a block of homozygosity on chromosome one, suggesting that the parents have a distant common ancestor. The novel homozygous frameshift mutation in *PAPPA2* (c.1927_1928insAT, p.D643fs25*) was highlighted as the likely causal variant, given the loss-of-function nature of the mutation and the role of the encoded protein, PAPP-A2, in cleaving IGFBPs. This variant was confirmed via Sanger sequencing to be homozygous in both affected individuals, heterozygous in both parents and one unaffected sibling (II.4), and absent in the second unaffected sibling (II.2) (Fig 2).

Due to the known consanguinity in family two, we assumed that the affected children were homozygous carriers of a rare autosomal recessive variant causal for their growth retardation. Genome-wide genotyping performed in the three affected children identified four large (> 6 MB) shared regions of homozygosity (Appendix Table S3). Whole-exome sequencing was performed on Patient II.3. Analysis focused on rare homozygous nonsynonymous variants within the shared regions of homozygosity. Two novel missense variants met these criteria, only one of which segregated in the family (Fig 2): a novel missense variant in *PAPPA2* (c.3098C > T, p.Ala1033Val).

Neither of the two *PAPPA2* variants was found in public databases representing more than 60,000 exomes and including at least 700 individuals from Spain. Notably, there are no individuals in these large datasets with homozygous loss-of-function variants of the *PAPPA2* gene. Thus, the genetic analyses of these two families identified two novel nonsynonymous mutations in *PAPPA2* as the likely causal variants.

   

**Table 1.  Anthropometric and biochemical data.**

| Subject ID | Sex | Gestational age (weeks) | Birth weight (kg) | Birth length (cm) | Age at examination (years) | Height (cm) | BMI (kg/m$^2$) | HC (SDS) |
|---|---|---|---|---|---|---|---|---|
| Family 1 | | | | | | | | |
| II.1 | F | 37 | 2.19 (−1.49) | 48.0 (−0.07) | 10.3 | 132.2 (−1.10) | 13.9 (−1.52) | −2.03 |
| II.2 | F | 39 | 3.35 (0.44) | 51.0 (0.94) | 7.8 | 129 (0.87) | 14.8 (−0.98) | +0.02 |
| II.3 | M | 35 | 1.99 (−1.27) | 45.0 (−0.55) | 5.7 | 110.4 (−0.96) | 13.4 (−2.03) | −2.19 |
| II.4 | F | 40 | 3.64 (0.96) | 50.0 (0.11) | 3.6 | 103.3 (1.62) | 15.6 (−0.44) | −0.98 |
| Family 2 | | | | | | | | |
| II.1 | F | 40 | 3.26 (−0.61) | 48.3 (−1.30) | 22.2 | 149.3 (−2.16) | 24.9 (0.76) | 0.45 |
| II.2 | F | 40 | 3.57 (−0.01) | 48.3 (−1.30) | 19.8 | 153.9 (−1.45) | 21.7 (−0.01) | 1.06 |
| II.3 | F | 41 | 2.58 (−2.16) | 45.7 (−2.75) | 18.6 | 138.4 (−3·81) | 20.9 (−0·19) | −2.03 |
| II.4[a] | M | 41 | 2.84 (−1.69) | 47.0 (−2.18) | 13.8 | 139.1 (−2.82) | 26.7 (1.61) | −1.60 |
| II.5 | M | 39.4 | 2.82 (−1.28) | 48.5 (−0.94) | 9.5 | 117.7 (−3.14) | 15.4 (−0.65) | −1.08 |

| | Total IGF-I (µg/l) | Total IGF-II (µg/l) | Bioactive IGF (µg/l) | Free IGF-I (µg/l) | BioIGF/Total IGF-I (%) | IGFBP-3 (µg/l) | IGFBP-5 (µg/l) | ALS (U/L) | PAPP-A2 (µg/l) |
|---|---|---|---|---|---|---|---|---|---|
| Family 1 | | | | | | | | | |
| II.1 | 957 | 1,198 | 0.87 | 1.39 | 0.09 | 5,912 | 997 | 3,745 | ND[b] |
| II.2 | 340 | 850 | 2.30 | 2.08 | 0.68 | 3,696 | 457 | 2,588 | 0.13 |
| II.3 | 882 | 1,264 | 1.54 | 0.27 | 0.17 | 4,850 | 853 | 3,625 | ND[b] |
| II.4 | 264 | 759 | 3.21 | 2.16 | 1.22 | 3,384 | 322 | 2,206 | 0.25 |
| Family 2 | | | | | | | | | |
| II.1 | 237 | 763 | 2.31 | 4.34 | 0.97 | 3,235 | 179 | 1,375 | 0.34 |
| II.2 | 490 | 629 | 3.90 | 8.36 | 0.80 | 3,007 | 187 | 1,412 | 0.44 |
| II.3[a] | 1,060 | 921 | 3.35 | 1.98 | 0.32 | 4,557 | 760 | 2,445 | 0.23 |
| II.4[c] | 935 | 914 | 2.70 | 3.31 | 0.29 | 4,792 | 981 | 2,959 | 0.42 |
| II.5 | 831 | 946 | 0.98 | 0.35 | 0.12 | 4,403 | 645 | 2,456 | 0.30 |
| Normal Values | | | | | | | | | |
| Tanner I | 91–225 | 495–911 | [d]Prepubertal 1.94 ± 0.15 | 1.58–3.15 | [e]median 1.23% (range 0.46–2.59%) | 2,206–4,200 | 211–707 | 753–2,634 | 0.16–2.69 |
| Tanner V | 270–617 | 560–794 | [d]Pubertal 2.93 ± 0.15 | 4.89–9.37 | | 2,796–5,280 | 293–1,023 | 1,260–4,030 | 0.23–0.80 |

Affected individuals are shaded in light gray. Unaffected individuals are not shaded. Values in parentheses indicate the standard deviation score (SDS) for anthropometric measures.
[a]This subject is an adult.
[b]Zero values in the affected patients are expected because of the frameshift mutation resulting in the absence of the target epitopes of the assay. HC: head circumference. BioIGF: Biologically active IGF.
[c]This subject was Tanner stage 2 at the time of examination and biochemical evaluation with 5-6 cc testes. ND: not detectable.
[d]Normal ranges for bioactive IGF are based on Sorensen et al (2015).
[e]The normal ratio of bioactive IGF to total IGF-I is based on 95 healthy subjects aged 7–25 years. The ratio was not associated with age.

***In vitro* functional effects of mutations in *PAPPA2***

PAPP-A2 is a metalloproteinase that specifically cleaves IGFBP-3 and -5 (Overgaard *et al*, 2001). To assess the consequences of our subjects' mutations on PAPP-A2 proteolytic function, recombinant variants were compared with wild-type (WT) PAPP-A2 following transfection of HEK 293 cells. Media from cells transfected with cDNA encoding the D643fs or A1033V variants showed no proteolytic activity toward IGFBP-3 or -5 under conditions where media from cells transfected with WT PAPP-A2 cDNA completely cleaved these substrates (Fig 3A). As expected,

none of the variants cleaved the PAPP-A substrate, IGFBP-4. Western blotting showed a complete lack of expression for the D643fs mutant (as would be expected with a severely truncated protein), while expression of A1033V was markedly reduced compared to WT PAPP-A2. Upon reduction of disulfide bonds, two bands appeared for A1033V, probably caused by partial intramolecular cleavage (Fig 3B). To further analyze the A1033V mutant, samples were adjusted for the reduced expression of this variant and, at equimolar concentrations (Fig 3C), it still showed no activity, suggesting that neither full-length A1033V nor proteolytically cleaved A1033V is capable of cleaving the IGFBPs.

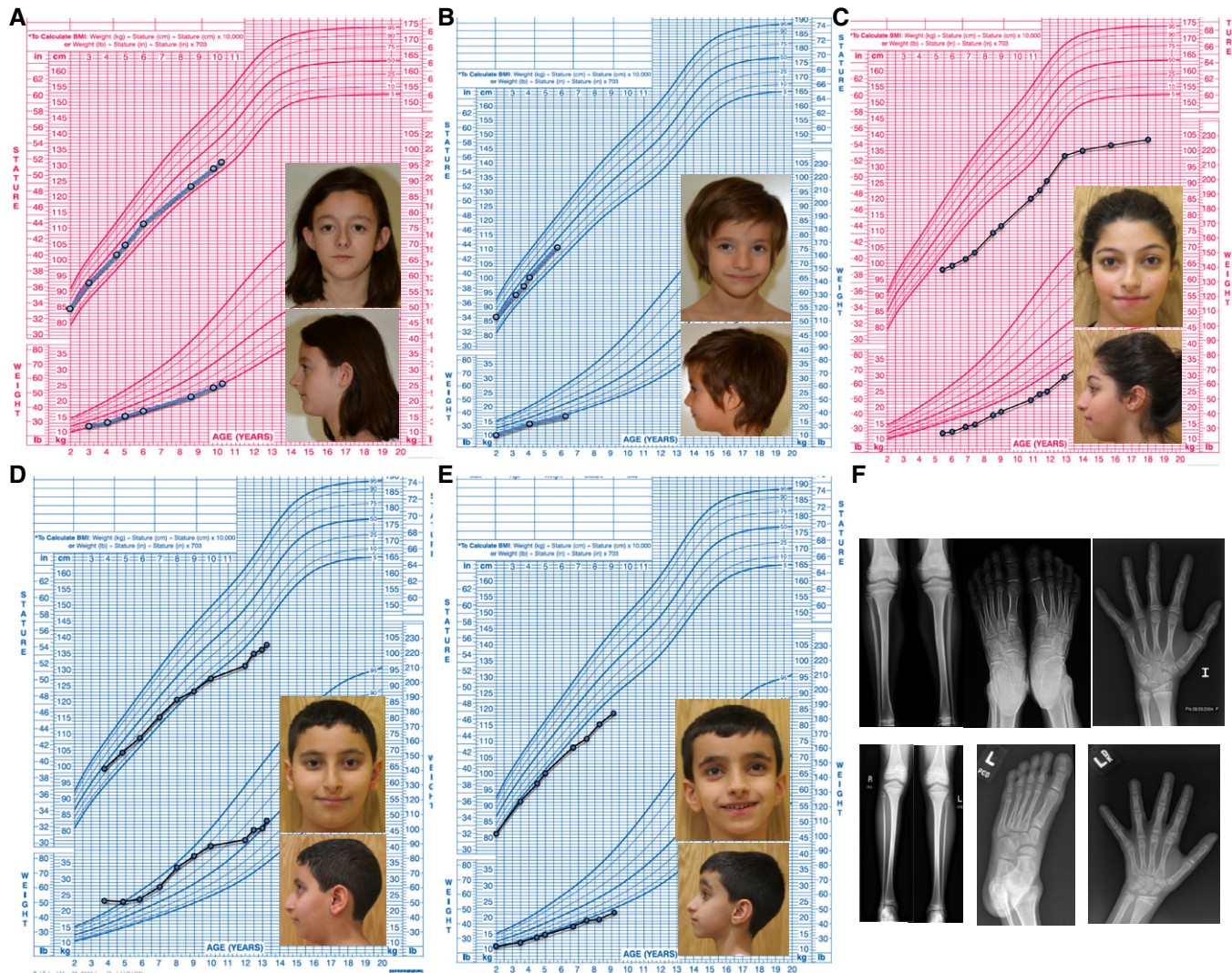

**Figure 1.   Growth charts and photographs of affected subjects.**

A  Family 1, Subject II.1.
B  Family 1, Subject II.3.
C  Family 2, Subject II.3.
D  Family 2, Subject II.4.
E  Family 2, Subject II.5.
F  Top Row—Representative X-rays showing narrow fibulae and elongated digits in Subject II.1 of Family 1. Bottom Row—Lower extremities and foot from Subject II.5 and hand from Subject II.4, both from Family 2.

### *In vivo* evaluation of IGF-I and IGFBPs

To further delineate the *in vivo* consequences of our subjects' mutations on IGF-I physiology, we measured serum PAPP-A2 concentrations. As expected, subjects in family one homozygous for the D463fs truncating mutation had no detectable PAPP-A2 (Table 1). In family two, PAPP-A2 concentrations were detectable at the low end of the normal range, but with no difference between the affected individuals and their heterozygous siblings.

Serum concentrations of total IGF-I were elevated above the normal range in all affected subjects and normal in their unaffected siblings (Table 1). IGFBP-3 concentrations were above the normal range in the four younger patients and in the upper normal range in the Tanner V subject of family two. ALS concentrations were elevated in three of five subjects and in the upper limits of normal in two. Serum IGFBP-5 concentrations were elevated in four of five patients. IGF-II concentrations were elevated in all five patients (Table 1). As PAPP-A2 cleaves IGFBP-3 and -5 thereby releasing IGF-I from the ternary complex, we hypothesized that fIGF-I concentrations would be decreased in the affected subjects, despite the elevated total IGF-I concentrations. Indeed, fIGF-I concentrations were low in four of five patients (Table 1), suggesting that they have a functional defect in the ability to liberate IGF-I from its binding partners. To substantiate this finding by an independent

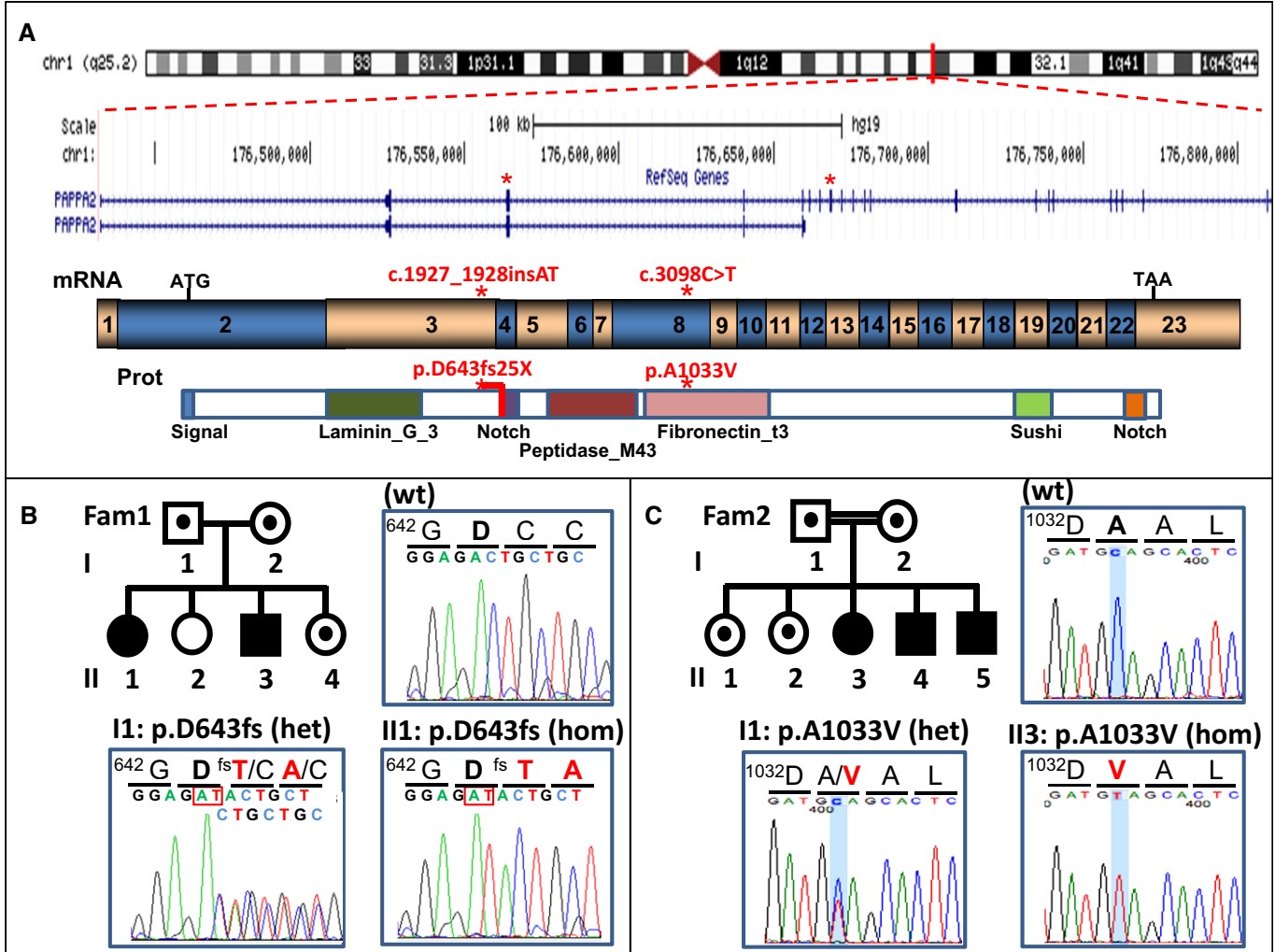

**Figure 2. *PAPPA2* gene and family pedigrees.**

A  Schematic representation of the genomic structure of the *PAPPA2* gene on chromosome band 1q24, the encoded mRNA and the protein with its relevant functional domains. Red stars indicate the location in the exons, mRNA and protein where the subjects' mutations are found.
B  Pedigree and representative Sanger sequencing trace of the mutation identified in family 1.
C  Pedigree and representative Sanger sequencing trace of the mutation identified in family 2.

method, the potential of the patients' serum to stimulate the IGF-IR *in vitro* was assessed. IGF bioactivity was lower in the three prepubertal patients compared to unaffected prepubertal subjects, with the percentage of bioactive IGF (bioactive IGF/total IGF-I) reduced in all affected patients (Table 1).

To further confirm that defects in PAPP-A2 function can cause this unique phenotype, we demonstrated that mice carrying a targeted deletion of *Pappa2* (Conover *et al*, 2011) also have markedly decreased fIGF-I concentrations, despite a > 50% increase in total IGF-I concentrations (Appendix Table S4).

Lastly, measurement of the amount of IGF-I bound in ternary complexes in the serum of subjects of family one showed a marked increase in IGF-I as part of the ternary complex in all affected subjects (Fig 4A). Moreover, in these patients, the overall capacity to form ternary complexes was greatly increased compared to prepubertal controls (Fig EV2A and B).

## Discussion

We present two families with a novel syndrome of progressive growth failure with markedly elevated serum concentrations of IGF-I, IGFBP-3 and -5 resulting from loss of PAPP-A2 function. PAPP-A2, a member of the pappalysin family of metzincin metalloproteinases, is closely related to PAPP-A, a protease that targets IGFBP-4 (Oxvig, 2015). While PAPP-A has been extensively studied due to its biomarker roles (Bayes-Genis *et al*, 2001; Malone *et al*, 2005), the function of PAPP-A2 in human physiology is largely unknown. Prior *in vitro* work has shown that PAPP-A2 specifically cleaves IGFBP-3 and -5 with no activity toward other IGFBPs (Overgaard *et al*, 2001). We demonstrate that loss of PAPP-A2 function leads to increased serum concentrations of IGFBP-3 and -5 resulting in increased total IGF-I concentrations. Total IGF-II is also elevated, consistent with the increased ALS and IGFBP-3 concentrations.

## A  Proteolytic activity in culture media of transfected cells

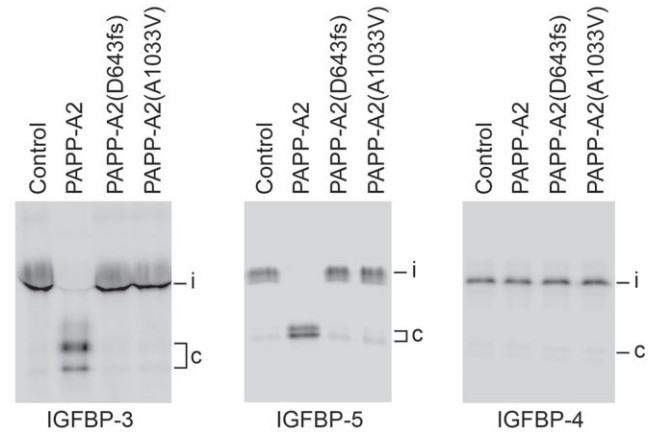

IGFBP-3          IGFBP-5          IGFBP-4

## B  PAPP-A2 Western blots

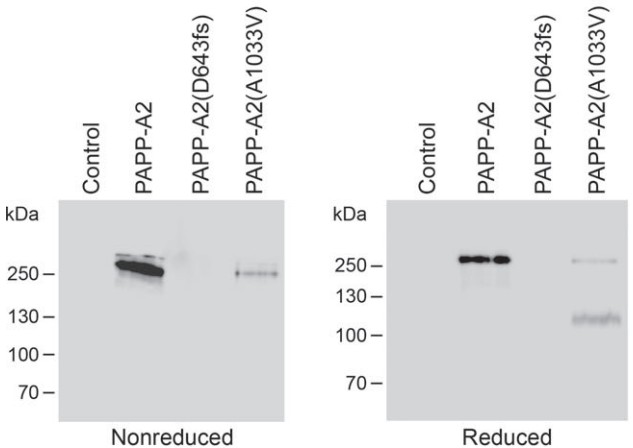

Nonreduced          Reduced

## C  Proteolytic activity of equimolar amounts

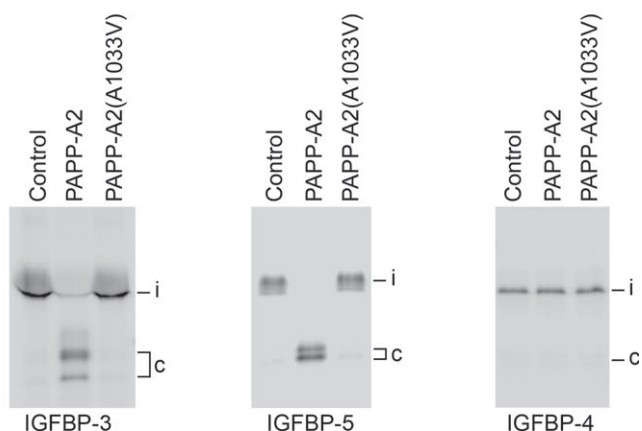

IGFBP-3          IGFBP-5          IGFBP-4

**Figure 3.  *In vitro* characterization of mutated PAPP-A2.**

A  Assessment of proteolytic activity present in culture media of human embryonic kidney 293T cells transfected with cDNA encoding PAPP-A2, PAPP-A2(D643fs) (family 1), or PAPP-A2(A1033V) (family 2). Radiolabeled substrates tested include IGFBP-3, IGFBP-4, and IGFBP-5. Cleavage was visualized by autoradiography following SDS–PAGE. Positions of intact IGFBPs (i) and cleavage products (c) are indicated.

B  Western blots following nonreducing or reducing SDS–PAGE of media from transfected cells, as indicated. Polyclonal antibodies raised against the N-terminal portion of PAPP-A2 were used for detection.

C  Comparison of proteolytic activity similar to the experiment of panel (A), except that equimolar concentrations of PAPP-A2 and PAPP-A2(A1033V) were used. The variant from Family 1, carrying a frameshift mutation N-terminal to the proteolytic domain of PAPP-A2, did not show any detectable expression. All gels and blots are representative of three independent experiments.

to an inability of the abnormal or deficient PAPP-A2 to liberate IGF-I from its binding partners resulting in decreased fIGF-I. The primary site of action for PAPP-A2 is unknown. Unlike PAPP-A, PAPP-A2 is not membrane bound and does not require the presence of IGF-I for its proteolytic function (Overgaard *et al*, 2001). We do not know whether the growth failure is due to a systemic effect or a defect in PAPP-A2 function at the tissue level, in particular at the growth plate. It is also possible that aspects of the patients' phenotype could be due to direct effects of the increased IGFBPs. Interestingly, GH is elevated in the patients, resulting in increased concentrations of all GH-dependent factors (IGF-I, IGFBP-3, IGFBP-5, and ALS). We hypothesize that the rise in GH is due to decreased negative feed-back consequent to low fIGF-I concentrations (Fig 4B). Thus, our study indicates that PAPP-A2 is a key regulator of human growth and IGF-I bioavailability by regulating the proportion of IGF-I that is either free or bound to its IGFBPs.

Mouse models of PAPP-A2 deficiency have striking similarities to our patients, as *Pappa2*[−/−] mice recapitulate the human biochemical phenotype with elevated total IGF-I and low fIGF-I concentrations. *Pappa2*[−/−] mice exhibit a trend toward decreased birth weight and significant postnatal growth retardation (Conover *et al*, 2011; Christians *et al*, 2013), which correlates with our findings as the patients were born mildly small for gestational age and exhibited the most prominent growth failure postnatally. Indeed, progressive postnatal growth retardation occurred in all affected individuals, becoming more prominent with age. While the range in the absolute heights of the affected patients is wide, subjects in family one are quite discordant from their target mid-parental height and it is yet to be seen if their growth failure becomes progressively worse with age, as seen in family two. PAPP-A2 is responsible for the proteolysis of IGFBP-5 during pregnancy resulting in increased fIGF-I (Yan *et al*, 2010), but how this affects fetal growth and development in normal individuals and in our subjects is unknown. Lack of PAPP-A2 led to decreased length of the mandible, skull, femur, pelvic girdle, and tailbone in mice (Christians *et al*, 2013) and severely reduced cranial cartilages in zebrafish (Kjaer-Sorensen *et al*, 2014). Our subjects had small chins, long thin fingers, thin femurs and moderate microcephaly. In the two subjects undergoing DXA, one showed mild osteoporosis and the other osteopenia. Mice with blunted IGF-I activity in osteoblasts show defects in bone mineralization during postnatal growth (Zhang *et al*, 2002). Transgenic mice overexpressing IGFBP-3 (Modric *et al*, 2001) have

However, this increase in total IGFs was not accompanied by an elevated ability of serum to activate IGF-IR *in vitro* (IGF bioactivity). We hypothesize that this relative decrease in IGF bioactivity is due

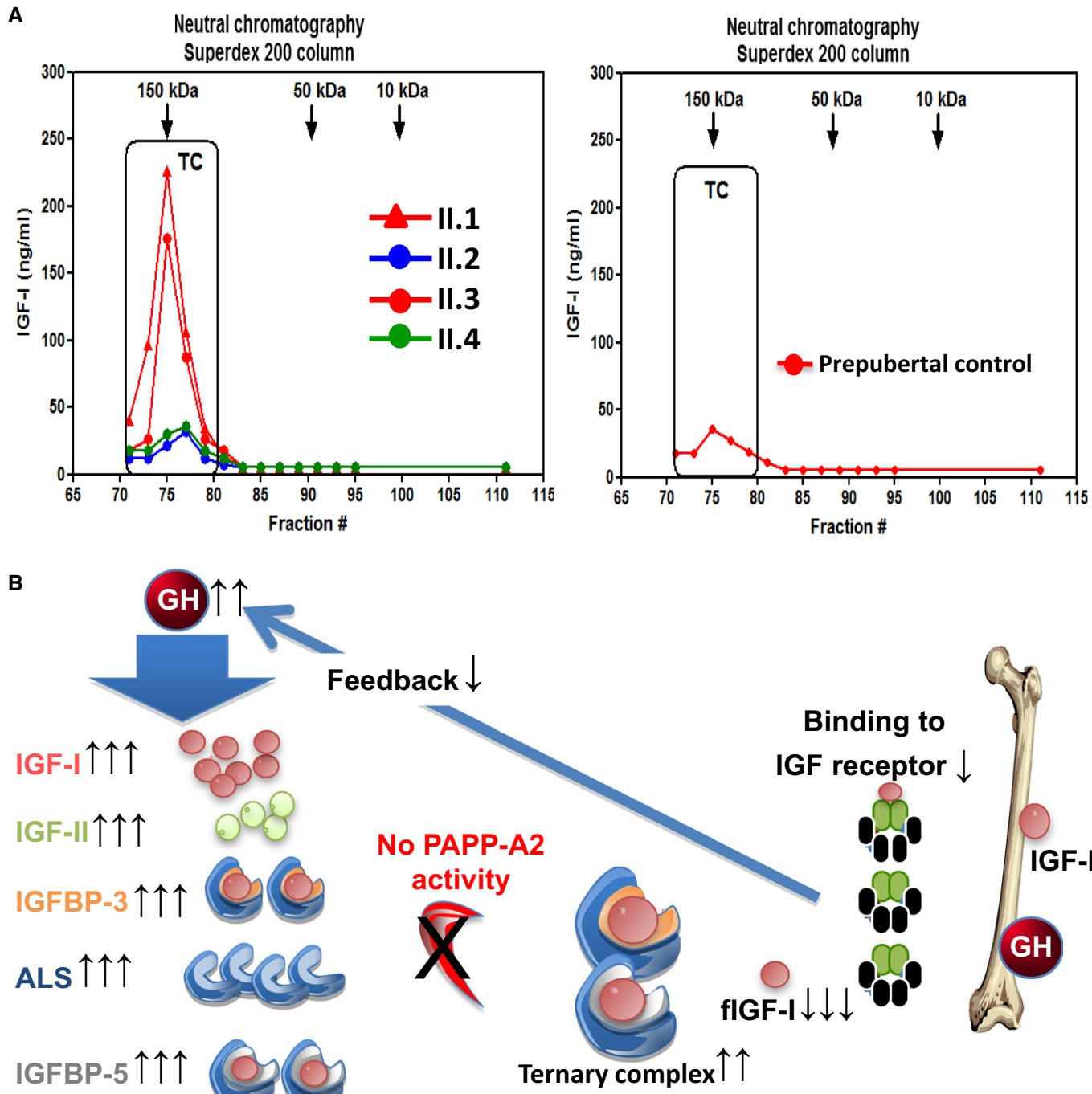

**Figure 4.   IGF-I neutral chromatography and proposed mechanism of action of PAPP-A2 deficiency.**

A   Concentrations of endogenous IGF-I in the different fractions obtained by neutral chromatography on a Superdex 200 column. Serum samples from the four siblings of family 1 were analyzed and compared to a nonrelated prepubertal control, as all siblings were prepubertal. The amount of IGF-I detected in the fractions corresponding to the ternary complex (TC) was markedly greater in the homozygous affected subjects (II.1 and II.3) compared to the unaffected siblings (II.2 and II.4) and the prepubertal control. No IGF-I was detected in the fractions corresponding to the binary complex (BC) or fIGF-I in any of the subjects. This is most likely due to the fact that under physiological conditions 80-85% of IGF-I circulates in the TC and after the separation procedure, the amount of IGF-I in the BC or free was below the level of detection of our IGF-I assay. These results clearly demonstrate that in the affected patients the amount of IGF-I circulating in the TC is markedly elevated.

B   Cartoon of hypothesized mechanism underlying growth failure due to lack of PAPP-A2. There is an increase in the formation of ternary complexes, due to decreased proteolysis subsequent to the lack of PAPP-A2 activity. This results in reduced concentrations of fIGF-I in serum and most likely at specific target tissues. This also decreases the negative feedback effect of IGF-I on growth hormone (GH) production, contributing to the increased circulating concentrations of GH. Increased serum GH concentrations cause increased IGF-I, IGFBP-3, and ALS concentrations. This increase in IGFBP-3 and ALS would then contribute to further ternary complex formation and an increase in total IGF-I and IGF-II concentrations.

modest pre- and postnatal growth retardation despite high IGF-I concentrations, presumably reflecting reduced fIGF-I, as seen in our patients. The mild fasting hyperinsulinemia in three of five patients is possibly due to the elevated GH concentrations.

The gene encoding PAPP-A2 was located in a quantitative trait locus for body size in mice (Christians et al, 2006) and PAPP-A2 was confirmed to be partially responsible for the effect of this locus (Christians et al, 2013). A SNP located in close proximity to human PAPPA2 (rs1325596) was recently identified in a large genome-wide association study as being correlated with height in the general population (Lango Allen et al, 2010; Wood et al, 2014). Thus, common genetic variants in PAPPA2 may have mild effects on growth, while rare loss-of-function variants, such as those seen in our subjects, have a more pronounced effect.

In conclusion, we have identified a new syndrome in two unrelated families with loss-of-function variants in PAPPA2 resulting in progressive growth failure, moderate microcephaly, thin long bones, a marked elevation of total IGF-I concentrations due to impaired proteolysis of IGFBP-3 and -5, and reduced fIGF-I. These patients represent the first human cases of diminished IGF-I bioavailability due to defects in IGFBP regulation, indicating that PAPP-A2 is a key regulator of IGF-I bioavailability and that deregulated IGFBP proteolysis can have relevant consequences in human physiology.

It remains to be seen whether postnatal growth in these children can be improved. One potential treatment includes human recombinant IGF-I. It is possible that increasing total IGF-I in the circulation could shift the equilibrium of the GH-IGF system such that there is also an increase in free/bioactive IGF-I, which could result in increased systemic growth. This hypothesis is now being tested as the prepubertal patients are under hrIGF-I therapy.

The results reported here also emphasize the potential importance of proteins that interact with PAPP-A2 (Kloverpris et al, 2013) that have been previously shown to affect human height (Lango Allen et al, 2010). Thus, patients currently labeled as idiopathic short stature with elevated IGF-I and IGFBP-3 concentrations could conceivably obtain a specific diagnosis after undergoing testing for mutations in PAPPA2, or possibly in proteins that regulate the activity of this protease. Indeed, the data presented here indicate that understanding PAPP-A2 regulation and function will have important implications in both the clinical diagnosis of growth retardation and other areas of IGF-I biology.

# Materials and Methods

Studies were approved by the Institutional Review Boards at the Hospital Infantil Universitario Niño Jesús and Cincinnati Children's Hospital Medical Center. Written informed consent was obtained from all subjects or their legal guardians and the experiments conformed to the principles set out in the WMA Declaration of Helsinki and the Department of Health and Human Services Belmont Report. Full consent was also obtained for the publication of the patients' pictures.

## Genetic analysis

In family two, homozygosity mapping analysis was implemented on genome-wide SNP data using PLINK (Purcell et al, 2007).

Whole-exome sequencing was performed in the probands of both families at each institution as previously described (Argente et al, 2014; de Bruin et al, 2015).

## Serum assays

ELISAs were used to measure serum concentrations of acid labile subunit (ALS; Mediagnost, Reutlingen, Germany), IGF-I, free IGF-I, IGFBP-3, IGFBP-5, and PAPP-A2 (Ansh Labs, Webster, TX, USA). All assays were performed according to the manufacturers' instructions. Briefly, in all determinations, after incubation of the serum samples in a microtiter plate previously coated with the corresponding antibody, the wells were washed and the enzyme conjugate was added. At the end of the incubation period and after washing, a substrate solution was added until the color developed adequately. Measurements were taken at 450 nm with an automatic microplate analyzer.

The PAPP-A2 ELISA employed is a two-site assay based on specific monoclonal antibodies. Epitope mapping of the two antibodies has been published earlier (Kloverpris et al, 2013). The catching antibody maps to the SCR3 module and the detecting antibody maps to the SCR1 module of PAPP-A2. All SCR modules are located C-terminal to the site of mutations.

Serum insulin concentrations were measured by an immunoradiometric assay from DIAsource ImmunoAssays (Louvain-la-Neuve, Belgium). Samples were dispensed, and a $^{125}$I-anti-insulin monoclonal antibody was added into the tubes coated with another anti-insulin monoclonal antibody. After incubation and washing, the bound fraction was counted in a gamma-counter.

In all assays, the intra- and inter-assay coefficients of variation were lower than 10%. Normal ranges were obtained by measuring concentrations in groups of Tanner Stage I or V healthy controls recruited from the clinic at the Hospital Infantil Universitario Niño Jesús ($n = 50$ in each group).

Bioactive IGF concentrations were measured by kinase receptor activation assay (KIRA) as previously described (Chen et al, 2003; Reinhard et al, 2013). Briefly, this assay measures the ability of IGF in serum to phosphorylate the IGF-I receptor (IGF-IR) in an in vitro-based model employing human embryonic cells transfected with cDNA of the human IGFR gene. The ability of serum to phosphorylate the IGF-IR in vitro is compared to a serial dilution of IGF-I. This assay also detects IGF-II and pro-IGF-II activation of the IGF-IR (cross-reactivity 12% and 2%, respectively), but the cross-reactivity of proinsulin and insulin is negligible (< 1%). Thus, the output of this assay is referred to as IGF bioactivity as both IGF-I and IGF-II can activate the IGF-IR. The detection limit of this assay is < 0.08 μg/l.

## Chromatography for endogenous IGF-I complex profiles

Levels of complex-bound endogenous IGF-I were assayed by neutral size-exclusion chromatography and the capacity to form such complexes as previously described (Domene et al, 2004). Briefly, to determine the IGF-I complex profile, we performed neutral chromatography of 300 μl of serum on Superdex 200. Two-ml fractions were collected and those corresponding to the ternary (six fractions), binary (seven fractions) and free IGF-I (one fraction) were collected and dried in a speed vacuum, reconstituted in 100 μl of

PBS buffer 0.05 M, NaCl 0.15 M, pH 7.4, BSA 0.5%, and IGF-I was extracted by alcohol-acid treatment and cryoprecipitation. The IGF-I concentration in each fraction was then determined by QLIA (Immulite, Siemens) according the manufacturer's instructions.

## Mutagenesis

Recombinant PAPP-A2 and variants thereof were generated by transfection of mammalian cells with WT or mutated cDNAs. Mutated proteins were compared to WT PAPP-A2 in their abilities to proteolytically cleave radiolabeled IGFBP-3, -4, and -5 and by Western blotting following nonreducing or reducing SDS–PAGE.

Plasmid constructs encoding variants of human PAPP-A2 were made by overlap extension PCR (Ho *et al*, 1989) using human *PAPPA2* cDNA contained in plasmid pPA2 (Overgaard *et al*, 2001) as a template. Outer primers were 5′-CTGTATGTGGATGGCACTC AGG-3′ and 5′-GCAACTAGAAGGCACAGTCGAG-3′. Overlapping mutated sets of internal primers were 5′-CTTTGACGACGGAGAT ACTGCTG-3′ and 5′-CAGCAGTATCTCCGTCGTCAAAG-3′ for mutant PAPP-A2(D643fs), and 5′-GAGGATAGAGATTGATGTAGCACTCCTG ACTTCTC-3′ and 5′-GAGAAGTCAGGAGTGCTACATCAATCTCTATC CTC-3′ for mutant PAPP-A2(A1033V). The mutated fragments were digested with XbaI and XhoI and swapped into pPA2 to generate plasmids pPA2(D643fs) and pPA2(A1033V), respectively. Plasmid DNA was prepared by GenElute HP Plasmid Miniprep Kit (Sigma). All constructs were verified by sequence analysis.

## Cell culture and transfection

Human embryonic kidney 293T cells (293tsA1609neo, ATCC CRL-3216) (Pear *et al*, 1993) were maintained in high-glucose DMEM supplemented with 10% fetal bovine serum, 2 mM glutamine, nonessential amino acids, and gentamicin (Invitrogen). The cells are routinely tested for mycoplasma contamination. For transient transfection, $6.0 \times 10^6$ cells were plated onto 10-cm dishes and transfected 18 h later by calcium phosphate coprecipitation using 10 μg plasmid DNA (Overgaard *et al*, 2000). In addition to plasmid constructs encoding PAPP-A2 or mutated variants, constructs encoding human IGFBP-3 (Laursen & Oxvig, 2005), human IGFBP-4 (Boldt *et al*, 2001), or human IGFBP-5 (Overgaard *et al*, 2001) were used. Culture supernatants were harvested 48 h post-transfection and cleared by centrifugation, or the cells were further cultured in serum-free medium (CD293, Invitrogen) to facilitate purification of the secreted proteins. Concentrations of WT PAPP-A2 in the media were determined by a two-site ELISA (Laursen *et al*, 2007).

## Assays for proteolytic activity

Purification of recombinant proteins (IGFBP-3, -4, and -5) was carried out by affinity chromatography on a 1 ml HisTrap HP column (GE Healthcare). Serum-free media were diluted 1:1 in 20 mM $NaH_2PO_4$, 150 mM NaCl, pH 7.4 (PBS) and loaded onto the column with a flow rate of 1 ml/min. The column was washed with 20 column volumes of 50 mM $NaH_2PO_4$, 1 M NaCl, 20 mM imidazole, 0.05% Tween-20, pH 7.4, followed by five column volumes of PBS. The proteins were eluted with 50 mM $NaH_2PO_4$, 300 mM imidazole, pH 7.4, and dialyzed against 20 mM HEPES, 150 mM NaCl, pH 7.4. Prior to iodination, the IGFBPs were further purified

by reversed-phase high-pressure liquid chromatography (RP-HPLC) on a Discovery BIO Wide Pore C5 column (4 × 250 mm, Sigma), as described (Gyrup & Oxvig, 2007). Protein purity was assessed by SDS–PAGE, and quantification of purified proteins was done by amino acid analysis. The purified IGFBPs were labeled with [125]I (Amersham Biosciences), and cleavage reactions were carried out as previously described for PAPP-A (Winn *et al*, 2009). In brief, media harvested from cells transfected with empty vector or human *PAPPA2* cDNA or variants thereof were diluted 1:25 and mixed with [125]I-IGFBP-5 (10 nM) in 50 mM Tris–HCl, 100 mM NaCl, 1 mM $CaCl_2$, pH 7.5. For wild-type PAPP-A2, this dilution corresponds to a concentration of 90 pM PAPP-A2 dimer. To assess activity toward IGFBP-3 and -4, culture media were diluted 1:10, corresponding to a PAPP-A2 concentration of 225 pM. Variants of all experiments in which the IGFBPs were preincubated with IGF-I (100 nM; GroPep Bioreagents) prior to the addition of PAPP-A2 were also carried out. Following 3 h of incubation at 37°C, the reactions were terminated by the addition of hot SDS–PAGE sample buffer supplemented with 25 mM EDTA. Substrate and cleavage products were separated by nonreducing SDS–PAGE and visualized by autoradiography using a storage phosphor screen (Molecular Dynamics) and a Typhoon imaging system (GE Healthcare). To assess specific activity toward the IGFBPs, cleavage reactions of 6 h were carried out using equimolar concentrations of PAPP-A2 wild-type and mutant PAPP-A2(A1033V).

## Western blot analysis

Proteins separated by nonreducing or reducing SDS–PAGE were blotted onto a PVDF membrane (Millipore), blocked with 2% Tween-20, and equilibrated in 50 mM Tris–HCl, 500 mM NaCl, 0.1% Tween-20, pH 9.0 (TST). The blots were incubated overnight at room temperature with an antiserum raised against the N-terminal laminin G-like module of PAPP-A2 diluted 1:10,000 in TST containing 0.5% fetal bovine serum (Winn *et al*, 2009). The blots were incubated for 1 h at room temperature with polyclonal swine anti-rabbit IgG-HRP (DAKO, P0217) diluted 1:2,000 in TST containing 0.5% fetal bovine serum. All washing between the steps was done with TST. The blots were developed using enhanced chemiluminescence (ECL Prime, GE Healthcare), and images were captured and analyzed using an ImageQuant LAS 4000 instrument (GE Healthcare). For quantification of mutant PAPP-A2 (A1033V), a standard curve was generated based on serial dilutions of wild-type PAPP-A2 of known concentration.

## *Pappa2* KO mice

All procedures involving mice complied with the standards stated in the "Guide for the Care and Use of Laboratory Animals" and were approved by the Institutional Animal Care and Use Committee of Mayo Clinic.

*Pappa2* KO mice are on a C57BL/6 background. They were generated at Lexicon Pharmaceuticals Inc. (The Woodlands, TX). *Pappa2* KO and wild-type littermates were housed in a standard pathogen-free mouse facility at Mayo Clinic with *ad libitum* access to food and water. Serum total and free IGF-I were measured in sixteen-week-old wild-type males ($n = 9$), wild-type females ($n = 12$), KO males ($n = 8$), and KO females ($n = 7$).

## The paper explained

### Problem

Syndromes marked by growth failure have been associated with mutations in multiple genes of the growth hormone (GH)/IGF-I axis. However, a large percentage of cases of familiar short stature continue to be classified as idiopathic. Identification of the specific cause underlying impaired longitudinal growth would most likely improve treatment protocols for the affected individual.

### Results

Here, we present two families with homozygous loss-of-function mutations in *PAPPA2*, a gene encoding pregnancy-associated plasma protein-A2 (PAPP-A2). PAPP-A2 is a protease highly specific for the cleavage of two of the six high-affinity IGF-binding proteins (IGFBP), IGFBP-3 and -5. In the circulation, most IGF-I is bound in ternary complexes in conjunction with the acid labile subunit (ALS) and IGFBP3 or IGFBP5 and must be released to act upon its target sites. The PAPP-A2 produced by both mutated genes lacks the ability to cleave either IGFBP3 or IGFBP5. This could explain the observed increase in the amount of IGF-I bound in ternary complexes and the decrease in free or bioactive IGF-I in these patients, which in turn most likely underlies their significantly decreased growth. Reduced levels of bioactive IGF-I would also reduce the ability of this hormone to act on the pituitary to reduce GH secretion. This would result in the increased GH secretion found in these patients, which in turn could explain the increased circulating levels of GH-dependent proteins such as IGFBP-3 and ALS. Moreover, mild modifications in bone mineral density and bone morphology were observed in these patients.

### Impact

This is a novel syndrome of growth retardation that presents with progressive growth failure with markedly elevated circulating IGF-I, IGF-II, IGFBP-3, IGFBP-5, and ALS, but decreased fIGF-I levels and IGF bioactivity. Analysis of PAPP-A2 in patients presenting with a similar phenotype will lead to a new specific diagnosis in a subset of patients now classified as idiopathic. Moreover, the data presented here indicate that understanding PAPP-A2 regulation and function will not only have important implications in the clinical diagnosis of growth retardation, but also in other areas of IGF-I biology.

## Micro-CT methods

Teeth were fixed in 4% paraformaldehyde and subjected to micro-CT analyses (Skyscan 1172, at 70 kV, 9.75 μm per pixel, 0.4 in degree rotation). Reconstruction of the raw images was performed using NRecon V1.4.0 software (Skyscan).

**Expanded View** for this article is available online.

## Acknowledgements

The authors would like to thank the patients and their families for their dedicated participation to this work. We would also like to thank Gopal Savjani, Ajay Kumar, and the entire team at Ansh Labs for their generous donation of assay kits and technical assistance. We thank John Kopchick for his helpful comments and insights and Sandra Canelles, Francisca Díaz, Raquel Flores, Melissa Andrew, Virginia Calley, and Laurie K. Bale for their assistance with sample processing and study coordination. Research reported in this publication was supported by Fondos de Investigación Sanitaria and fondos FEDER (Grants PI100747 and PI1302195 to JA, PI1302481 to LAPJ), Ministerio de Ciencia e Innovación (Grants BFU2011–27492 and BFU2014-51836-C2-2-R to JAC), Centro de Investigación Biomédica en Red Fisiopatología de Obesidad y Nutrición (CIBEROBN), Instituto de Salud Carlos III (JA), Fundación Endocrinología y Nutrición (JA), the Catalan Government (2014SGR1468 and ICREA Acadèmica to LAPJ), the Eunice Kennedy Shriver National Institute of Child Health & Human Development of the National Institutes of Health (Award Number K23HD07335 to AD), The Danish Council for Independent Research (FNU), and the Novo Nordisk Foundation (CO). A CIBER for Rare Diseases (CIBERER) fellowship supported CSJ. The funding sources had no role in the design or interpretation of the results or in the decision to submit these data for publication. The content is solely the responsibility of the authors and does not necessarily represent the official views of the National Institutes of Health.

## Author contributions

AD and JA designed and conducted the experiments, analyzed the data, phenotyped the patients, and wrote the manuscript. VB, HMD, SK, CSJ, FH, HGJ, CAC, JF, SY, VH, JAC, CO, and LAPJ conducted the experiments, analyzed the data, and reviewed the manuscript. MTMC, GÁM, JP, VD, and RM phenotyped the patients, analyzed the data, and reviewed the manuscript. RGR analyzed the data and reviewed the manuscript.

## Conflict of interest

Drs. Dauber, Hwa, and Argente have a patent application pending entitled "Enzyme replacement therapy of PAPP-A2 (Application No: 62/126,831)" relevant to this work. Otherwise, the authors have no conflict of interest to declare.

## For more information

http://exac.broadinstitute.org/
http://evs.gs.washington.edu/EVS/
http://geevs.crg.eu/
http://bioinfo.cipf.es/ apps-beta/exome-server/beta/

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
