## [Review Process File · EMBO Molecular Medicine]

Mutations in pregnancy-associated plasma protein A2 cause short stature due to low IGF-I availability

Andrew Dauber, María T. Muñoz-Calvo, Vicente Barrios, Horacio M. Domené, Soren Kloverpris, Clara Serra-Juhé, Vardhini Desikan, Jesús Pozo, Radhika Muzumdar, Gabriel Á. Martos-Moreno, Federico Hawkins, Héctor G. Jasper, Cheryl A. Conover, Jan Frystyk, Shoshana Yakar, Vivian Hwa, Julie A. Chowen, Claus Oxvig, Ron G. Rosenfeld, Luis A. Pérez-Jurado, Jesús Argente

Corresponding author: Jesús Argente, Hospital Infantil Universitario Niño Jesús, Universidad Autónoma de Madrid

Review timeline:	Submission date:	27 November 2015
	Editorial Decision:	11 January 2016
	Revision received:	18 January 2016
	Accepted:	25 January 2016

Transaction Report:

Editor: Céline Carret

1st Editorial Decision

11 January 2016

Thank you for the submission of your revised manuscript to EMBO Molecular Medicine and please accept my sincere apologies for the delay in getting back to you, due to the festive season. We have now received the enclosed reports from the referees who assessed your article. As you will see, the reviewers are globally supportive and I am pleased to inform you that we will be able to accept your manuscript pending the following final amendments:

1) please address all minor issues highlighted by the reviewers and provide a point-by-point response to their comments (as word file).

Please submit your revised manuscript as soon as possible.

***** Reviewer's comments *****

Referee #1 (Comments on Novelty/Model System):

The finding of a new syndrome due to PAPP2 mutations is an important contribution to pediatric endocrinology and growth homeostasis. It underscores the physiological importance of a new effector of IGF1 transport and raises novel questions about growth regulatory pathways. Genetics

are strongly supported by clear functional demonstration of biochemical defects.

Referee #1 (Remarks):

The finding of a new syndrome due to PAPP2 mutations by Dauber et al is an important contribution to the physiology of growth and the mechanisms that can alter IGF1 transport and effects on growth plates. Genetic findings are clear. functional experiments that show the biochemical consequences of the mutations in vitro and in vivo provide a strong and undisputable support to the genotype phenotype relationship. The paper is well written and concise.

I would only suggest micro modifications.

Page 8: replace "extremely elevated" by a more appropriate wording

Page 10: "This is a unique....in human short stature" is not true: children with PROP1 mutations revealing late in childhood, as well as children with mild forms of SHOX mutations may well show this kind of growth trajectories. I would simply skip this sentence.

Discussion: The authors should give their opinion and possibly their experience about treatment issues, notably the use of recombinant IGF1.

Referee #2 (Comments on Novelty/Model System):

This is generally a well-written manuscript describing a novel molecular mechanism that accounts for an unusual short stature phenotype in 2 distinct human pedigrees. Biochemical data have been reproduced satisfactorily in a pre-existing mouse model. Overall this manuscript has significantly advanced knowledge in the field and is worthy of publication.

Referee #2 (Remarks):

This is an interesting and well-written manuscript that details the documentation of homozygous mutations in PAPP-A2 in 2 pedigrees with short stature. The authors have provided good genetic evidence as well as functional analysis of the mutations that implicate the gene in the aetiology of the short stature. The mechanism is also plausible - mutations in PAPP-A2 interfere with cleavage of IGFBP3 and IGFBP5 and hence although the total IGF1 concentrations are high, the concentration of fIGF1 is low. This would then lead to impaired growth. I would suggest that the following changes be considered:

1. Introduction: I would suggest that the authors replace the word "correct human growth" by "optimal human growth".
2. The authors need to state that HEK293 cells were used in the Methods.
3. In the Results section, at what time was the secretion of GH assessed - was it during the night?
4. The growth velocity in Figure 1A and 1B does not appear to be very slow, whereas in Pedigree 2, the growth velocity appears to be much slower. Additionally, in Figure 1D, the growth rate is slow with an increased weight gain. This could be associated with the hyperinsulinaemia - were serial measurements made of insulin concentrations in patient II.4 in Pedigree 2?
5. Suggest that "concentrations" should be used instead of "levels", and decimal ages should be used throughout.
6. It would be important to document the fact that the SGA is relative - and pedigree specific. Affected children had lower birth weights than unaffecteds.
7. The manuscript would be enhanced by the addition of DXA and skeletal survey data in Pedigree 2. Were there any tooth abnormalities in Pedigree 2.
8. Could the authors discuss a potential role for recombinant human IGF1 therapy?
9. It would appear that the growth failure appears to be mainly pubertal - could the authors comment on this phenomenon?

We would like to thank both referees for their positive comments and constructive suggestions that helped us to improve the quality of this manuscript.

Referee #1 (Remarks):

The finding of a new syndrome due to PAPP2 mutations by Dauber et al is an important contribution to the physiology of growth and the mechanisms that can alter IGF1 transport and effects on growth plates. Genetic findings are clear. functional experiments that show the biochemical consequences of the mutations in vitro and in vivo provide a strong and undisputable support to the genotype phenotype relationship. The paper is well written and concise.

I would only suggest micro modifications.

Page 8: replace "extremely elevated" by a more appropriate wording

Response: We have modified this statement as suggested.

Page 10: "This is a unique...in human short stature" is not true: children with PROP1 mutations revealing late in childhood, as well as children with mild forms of SHOX mutations may well show this kind of growth trajectories. I would simply skip this sentence.

Response: We have removed this statement as suggested.

Discussion: The authors should give their opinion and possibly their experience about treatment issues, notably the use of recombinant IGF1.

Response: We have included a statement regarding the use of recombinant IGF-I in these patients (Page 11).

Referee #2 (Comments on Novelty/Model System):

This is generally a well-written manuscript describing a novel molecular mechanism that accounts for an unusual short stature phenotype in 2 distinct human pedigrees. Biochemical data have been reproduced satisfactorily in a pre-existing mouse model. Overall this manuscript has significantly advanced knowledge in the field and is worthy of publication.

Referee #2 (Remarks):

This is an interesting and well-written manuscript that details the documentation of homozygous mutations in PAPP-A2 in 2 pedigrees with short stature. The authors have provided good genetic evidence as well as functional analysis of the mutations that implicate the gene in the aetiology of the short stature. The mechanism is also plausible - mutations in PAPP-A2 interfere with cleavage of IGFBP3 and IGFBP5 and hence although the total IGF1 concentrations are high, the concentration of fIGF1 is low. This would then lead to impaired growth. I would suggest that the following changes be considered:

1. Introduction: I would suggest that the authors replace the word "correct human growth" by "optimal human growth".

Response: This has been changed as suggested.

2. The authors need to state that HEK293 cells were used in the Methods.

Response: This information has been included in the Materials and Methods (Page 15).

3. In the Results section, at what time was the secretion of GH assessed - was it during the night?

Response: This analysis was performed during the day. This is now included in the methods section.

4. The growth velocity in Figure 1A and 1B does not appear to be very slow, whereas in Pedigree 2, the growth velocity appears to be much slower. Additionally, in Figure 1D, the growth rate is slow with an increased weight gain. This could be associated with the hyperinsulinaemia - were serial measurements made of insulin concentrations in patient II.4 in Pedigree 2?

Response: There is a clear deterioration of the growth velocity in both patients of Family 1 in relationship to their target height (P50-75). Regarding the possibility of hyperinsulinaemia in patient II.4 in Family 2, this is of course a good observation. We also believe that this is a possibility. However, serial measurements of insulin concentrations have not been performed in this patient.

5. Suggest that "concentrations" should be used instead of "levels", and decimal ages should be used throughout.

Response: We have made the suggested changes.

6. It would be important to document the fact that the SGA is relative - and pedigree specific. Affected children had lower birth weights than unaffected.

Response: We have clarified this in the manuscript. None of the patients were classified as SGA, according to the International Consensus. The male patient in pedigree 1 was born premature, but adequate for gestational age, followed by normal catch-up growth. This can be appreciated in Annex Figure 1.

7. The manuscript would be enhanced by the addition of DXA and skeletal survey data in Pedigree 2. Were there any tooth abnormalities in Pedigree 2.

Response: Unfortunately we do not have DXA data in Pedigree 2 at the moment of diagnosis. In contrast, the skeletal survey data was performed and some of this data are represented in Figure 1.

We did not have the opportunity to study any tooth in this family as of yet.

8. Could the authors discuss a potential role for recombinant human IGF1 therapy?

Response: This has been included in the Discussion section (Page 11). Patients in Family 1 are currently under human recombinant IGFI therapy.

9. It would appear that the growth failure appears to be mainly pubertal - could the authors comment on this phenomenon?

Response: The auxological data indicate that there is a slow deterioration of growth velocity prepubertally. Indeed, it appears that this deterioration of growth velocity is accentuated pubertally. However, as all patients have not yet reached puberty, it is difficult to make a statement in this regard. Hence, we would prefer not to make this generalization until all patients have reached puberty.

Corresponding Author Name: Jesús Argente
 Journal Submitted to: EMBO Molecular Medicine
 Manuscript Number: EMM-2015-06106